# Feasibility and Sensitivity of Saliva GeneXpert MTB/RIF Ultra for Tuberculosis Diagnosis in Adults in Uganda

Patrick Byanyima,[a] Sylvia Kaswabuli,[a] Emmanuel Musisi,[b] Catherine Nabakiibi,[a] Josephine Zawedde,[a] Ingvar Sanyu,[a] Abdul Sessolo,[a] Alfred Andama,[a,e] William Worodria,[a,e] Laurence Huang,[c,d] J. Lucian Davis[f,g]

[a]Infectious Diseases Research Collaboration, Kampala, Uganda
[b]Division of Infection and Global Health, School of Medicine, University of St. Andrews, United Kingdom
[c]Division of Pulmonary and Critical Care Medicine, University of California San Francisco, San Francisco, California, USA
[d]Division of HIV, Infectious Diseases, and Global Medicine, University of California San Francisco, San Francisco, California, USA
[e]Department of Internal Medicine, Makerere University College of Health Sciences, Kampala, Uganda
[f]Department of Epidemiology of Microbial Diseases, Yale School of Public Health, New Haven, Connecticut, USA
[g]Pulmonary, Critical Care, and Sleep Medicine Section, Yale School of Medicine, New Haven, Connecticut, USA

**ABSTRACT** The objective of this prospective observational study carried out at China-Uganda Friendship Hospital-Naguru in Kampala, Uganda, was to determine the performance of GeneXpert MTB/RIF Ultra (Xpert Ultra) molecular testing on saliva for active tuberculosis (TB) disease among consecutive adults undergoing TB diagnostic evaluation who were Xpert Ultra positive on sputum. We calculated sensitivity to determine TB diagnostic performance in comparison to a composite reference standard of *Mycobacterium tuberculosis* liquid and solid cultures on two spot sputum specimens. Xpert Ultra on a single saliva sample had a sensitivity of 90% (95% confidence interval [CI], 81 to 95%) relative to the composite sputum culture-based reference standard, similar to the composite sensitivity of 87% (95% CI, 77 to 94%) for fluorescence microscopy (FM) for acid-fast bacilli on two sputum smears. The sensitivity of salivary Xpert Ultra was 24% lower (95% CI for difference, 2 to 48%; $P = 0.003$) among persons living with HIV (71%; 95% CI, 44 to 90%) than among persons living without HIV (95%; 95% CI, 86 to 99%) and 46% higher (95% CI, 14 to 77%; $P < 0.0001$) among FM-positive (96%; 95% CI, 87 to 99%) than among FM-negative (50%; 95% CI, 19 to 81%) patients. The semiquantitative Xpert Ultra grade was systematically higher in sputum than in a paired saliva sample from the same patient. In conclusion, molecular testing of saliva for active TB diagnosis was feasible and almost as sensitive as molecular testing of sputum in a high TB burden setting.

**IMPORTANCE** Tuberculosis is among the leading causes of morbidity and mortality worldwide, in large part because >3 million people go undiagnosed and untreated each year. Sputum has been the mainstay for TB diagnosis for over a century but can be difficult for patients to produce. In addition, the vigorous coughing required during sputum collection can lead to infection of nearby individuals and health workers. In this case-only study, applying the ultra-sensitive GeneXpert MTB/RIF Ultra molecular diagnostic assay to saliva detected 90% of culture-confirmed TB cases among 81 adults who were undergoing TB evaluation at the outpatient department of a general hospital in Uganda and tested sputum GeneXpert MTB/RIF Ultra positive. These results suggest that saliva may be a feasible and sensitive alternative to sputum for TB diagnosis, thereby meeting two key metrics proposed by the World Health Organization in its target performance profile for a nonsputum test for TB.

**KEYWORDS** HIV/AIDS, nucleic acid amplification techniques, Uganda, diagnosis, feasibility, saliva, sensitivity, tuberculosis

Address correspondence to J. Lucian Davis, lucian.davis@yale.edu.

The authors declare no conflict of interest.

Over the last quarter century, improvements in diagnosis and treatment of people with tuberculosis (TB) have gradually reduced mortality, but large gaps in detection and treatment persist that contribute to substantial ongoing morbidity and mortality (1). Among several available strategies to facilitate rapid, same-day diagnosis of TB, testing sputum with the GeneXpert MTB/RIF Ultra (Xpert Ultra) molecular assay (2, 3) is the most sensitive and readily available approach. Unfortunately, there are several operational challenges associated with collecting sputum for diagnosis of pulmonary TB. First, coughing during sputum expectoration or sputum induction generates aerosols that may facilitate TB transmission (4). Second, some individuals are unable to produce sputum, including young children, those with dry cough, and the severely ill or severely debilitated. Given these limitations of sputum for TB diagnosis, in 2014, the World Health Organization (WHO) issued guidelines for developers of a future nonsputum test for active TB diagnosis, including a target product profile suggesting that an acceptable nonsputum test should have a minimum diagnostic accuracy similar to the previous generation GeneXpert TB/RIF assay on sputum smear-negative individuals (i.e., sensitivity $\geq$ 68%, specificity $\geq$ 98%) (5).

One nonsputum sample type with great promise for diagnosis of pulmonary TB is saliva, which is easy to collect without coughing, thereby reducing the risk of aerosol generation and TB transmission during diagnostic evaluation for TB. Although Stop TB Partnership guidelines discourage collection of salivary sputum samples because they have a lower diagnostic yield for acid-fast bacilli (AFB) when assayed by smear microscopy or mycobacterial culture, their diagnostic yield appears much more promising when assayed using TB molecular testing. In a previous study of 1,782 smear-negative adults undergoing evaluation for active TB, we found that salivary sputum had a substantially higher diagnostic yield and sensitivity (66%; 95% confidence interval [CI], 53 to 77%) for culture-positive TB than mucoid sputum (52%; 95% CI, 46 to 58%) and other sputum types (i.e., mucopurulent, bloody), suggesting that saliva might add diagnostic value over sputum alone (6). Using a different sampling technique, oral swab analysis, Wood and colleagues confirmed that TB is indeed present in the oral cavity when they reported that nylon swabs of the tongue and mouth assayed via insertion sequence IS*6110* PCR detected 18 of 20 (90%) South African patients with TB confirmed via the pre-Ultra GeneXpert MTB/RIF assay on sputum (7). A larger and more recent study from Uganda of oral swab analysis with Xpert Ultra in 183 adults with possible TB showed a slightly lower sensitivity of 78% (95% CI, 64 to 88%) relative to relative to a sputum reference standard and an outstanding specificity of 100% (95% CI, 97 to 100%) (8). Saliva is also now widely used for molecular diagnosis of COVID-19, where it has high sensitivity, even among patients without symptoms (9). Using saliva as a diagnostic specimen in the COVID-19 context has been shown to reduce aerosol exposure for health care workers and eliminate the need for personal protective equipment because it is self-collected (10). This raises the possibility that saliva could be used as a stand-alone TB diagnostic if paired with ultrasensitive, next-generation molecular tests such as Xpert Ultra, whose limit of detection (15.6 CFU/mL) (11) approximates that of mycobacterial culture (<10 CFU/mL) (12). Thus, the aim of this study was to assess the performance of Xpert Ultra on saliva among patients being evaluated for TB who were sputum Xpert Ultra positive.

## RESULTS

**Study population.** Among 153 participants enrolled in the parent study between June 2018 and June 2019, we excluded 40 who were sputum Xpert Ultra negative (only 2 of whom were sputum culture positive), 15 who did not have sputum Xpert Ultra performed (13 of whom were TB positive by both fluorescence microscopy and mycobacterial culture), 16 who did not provide a saliva specimen, and 1 participant who had an indeterminate culture result (Fig. 1), leaving 81 participants for inclusion in the analysis. Eighty-two of the 98 patients who met eligibility criteria successfully provided saliva, and there were no adverse events during specimen collection. The median age of participants was 30 years (interquartile range, 24 to 38), and 50 (62%) were men. Eighteen (22%) were persons living with HIV, with a median CD4 cell count of 90 cells/$\mu$L (interquartile range, 49 to 234), and only 7 of the 18 (39%) were taking antiretroviral therapy at enrollment. Ten (12%) of the 81 participants

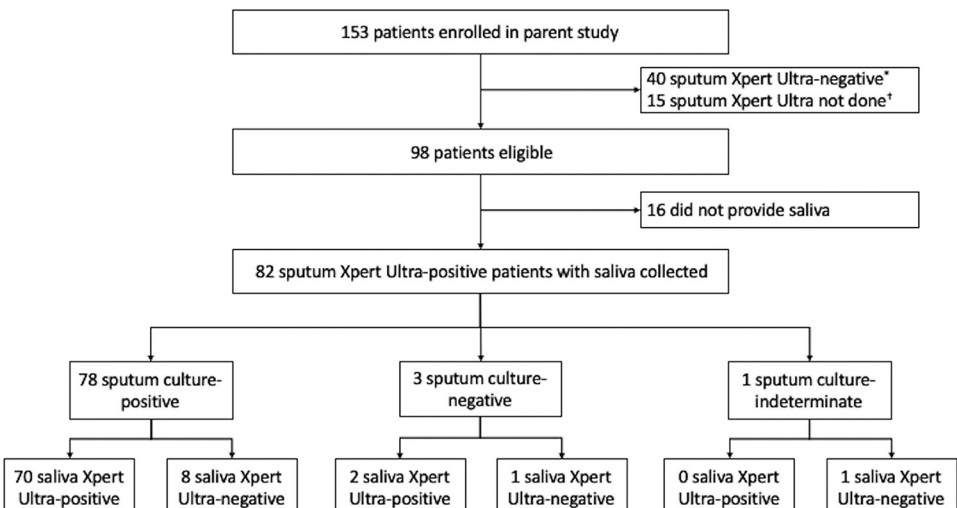

**FIG 1** Flow diagram showing study enrollment and TB reference standard results. *, Thirty-eight out of 40 patients who were excluded for being sputum Xpert Ultra negative were also composite sputum mycobacterial culture negative, leaving only two sputum Xpert Ultra-negative patients who were culture positive. †, Thirteen out of 15 patients excluded because sputum Xpert Ultra testing was not done were sputum microscopy positive and sputum culture positive, while two were culture indeterminate.

reported daily smoking, and 23 (28%) reported daily alcohol use. Seventeen (21%) reported cough for greater than 2 weeks, while 71 (88%) reported subjective fever within the past 7 days. Seventy-five (93%) reported weight loss, including 46 (57%) with weight loss of ≥5 kg. Fourteen (17%) reported no ambulatory limitation, 47 (58%) were mildly limited with ambulation, 12 (15%) were moderately affected and sometimes confined to bed, and 8 (10%) were severely affected and usually confined to bed. Sixty-seven patients (83%) were AFB smear positive, including 30 (37%) with an AFB microscopy smear grade of 3+, 18 (22%) with a grade of 2+, 9 (11%) with a grade of 1+, and 10 (12%) had 1 to 9 AFB seen per 100 high-powered fields. Thirteen (16%) were AFB smear-negative, and one (1%) was missing an AFB smear microscopy result (Table 1).

**Diagnostic performance.** Seventy-eight patients were confirmed sputum *M. tuberculosis* culture positive on liquid and/or solid media, while three were sputum *M. tuberculosis* culture negative. Seventy of the 78 patients with sputum culture-confirmed TB were salivary Xpert Ultra positive, giving an overall sensitivity of 90% (95% CI, 81 to 95%) relative to the composite sputum culture-based reference standard. Seventy-two of the 81 sputum Xpert Ultra-positive patients were salivary Xpert Ultra positive, giving an overall sensitivity of Xpert Ultra on saliva of 89% (95% CI, 80 to 95%) relative to the sputum molecular reference standard. In comparison, the composite sensitivity of fluorescence smear microscopy (FM) on two sputa was 87% (95% CI, 77 to 94%) relative to the sputum culture-based reference standard and 84% (95% CI, 74 to 91%) relative to the sputum molecular reference standard. The sensitivity of sputum Xpert Ultra was 100% (95% CI, 95 to 100%) relative to the sputum culture-based reference standard and 100% (95% CI, 96 to 100%) relative to the sputum molecular reference standard.

The sensitivity of Xpert Ultra on saliva among persons living with HIV (71%; 95% CI, 44 to 90%) was 24% lower (95% CI for difference, 2 to 48%; *P* = 0.003) than among persons living without HIV (95%; 95% CI, 86 to 99%), relative to a sputum culture-based reference standard. The sensitivity of Xpert Ultra on saliva was 46% higher (95% CI for difference, 14 to 77%; *P* < 0.0001) among sputum microscopy-positive participants (96%; 95% CI, 87 to 99%) than among sputum microscopy-negative patients (50%; 95% CI, 19 to 81%), again relative to a sputum culture-based reference standard.

We also compared the semiquantitative results for Xpert Ultra on saliva and sputum, as shown in Table 2. Overall, the semiquantitative Xpert Ultra grade was higher in sputum samples than in paired saliva samples collected from the same patient: 56 of 72 (78%) sputum

**TABLE 1** Demographic and clinical characteristics of patients[c]

| Characteristic ($n = 81$) | Value (no. [%])[a] |
|---|---|
| Age (median [Q1, Q3] [yrs]) | 30 (24, 38) |
| Men | 50 (62) |
| Inpatients | 7 (9) |
| Persons living with HIV | 18 (22) |
| CD4[+] T-cell count (median cells/mL [Q1, Q3]) | 90 (49, 234) |
| Taking antiretroviral therapy at enrollment | 7 (39) |
| Cigarette smoking, daily | 10 (12) |
| Alcohol use, daily | 23 (28) |
| Cough of any duration | 81 (100) |
| Cough for ≥14 days | 17 (21) |
| Fever within the past 7 days | 71 (88) |
| Weight loss | 75 (93) |
| Difficulty in breathing within the past 7 days | 46 (57) |
| | |
| Clinical status | |
| Unaffected, ambulatory | 14 (17) |
| Mildly affected, ambulatory | 47 (58) |
| Moderately affected, in bed ≤50% of the day | 12 (15) |
| Severely affected, in bed >50% of the day | 8 (10) |
| Completely disabled, bedbound | 0 |
| | |
| Sputum AFB microscopy grade[b] | |
| 3+ | 30 (37) |
| 2+ | 18 (22) |
| 1+ | 9 (11) |
| 1–9 per 100 hpf | 10 (12) |
| Negative | 13 (16) |

[a]Number and percent are shown unless otherwise specified.
[b]One result missing.
[c]AFB, acid-fast bacilli; hpf, high-powered fields; Q1, 25% quartile; Q3, 75% quartile; TB, tuberculosis.

samples were of either medium ($n = 22$) or high ($n = 34$) semiquantitative grade, whereas only 14 of 72 (19%) saliva samples were of either medium ($n = 10$) or high ($n = 4$) grade, an indication that the mycobacterial load in saliva specimens was low overall. There was no difference in semiquantitative results by smear microscopy result ($P = 0.52$) or by HIV status ($P = 0.39$).

## DISCUSSION

In a prospective observational study of consecutive sputum Xpert Ultra-positive TB patients in a high-burden setting, we showed that diagnosis of TB using Xpert Ultra on

**TABLE 2** Within-individual comparisons of semiquantitative Xpert MTB/RIF Ultra results between sputum and saliva ($n = 81$)[a]

| Sputum Xpert Results | Saliva Xpert Results | | | | | |
|---|---|---|---|---|---|---|
| | Negative | Trace | Very Low | Low | Moderate | High |
| Negative | 0 | 0 | 0 | 0 | 0 | 0 |
| Trace | 0 | 0 | 0 | 0 | 0 | 0 |
| Very Low | 4 | 0 | 0 | 1 | 0 | 0 |
| Low | 2 | 2 | 5 | 6 | 0 | 2 |
| Moderate | 3 | 0 | 4 | 15 | 2 | 1 |
| High | 0 | 2 | 3 | 20 | 8 | 1 |

[a]Shading intensity is proportional to the frequency of paired results by semiquantitative grade across the two sample types.

saliva is feasible and had a high sensitivity relative to a rigorously defined reference standard of composite sputum mycobacterial culture. This finding has significant implications for the diagnosis of TB and potentially also for TB infection control. Using sputum specimens for TB diagnosis poses a number of challenges since some individuals, such as young children and those with nonproductive cough, find expectoration challenging, and the associated generation of sputum aerosols poses an infection control risk for health care workers and nearby patients (13). The development of novel testing strategies that employ nonsputum samples for TB has been identified as a priority by the WHO, and the sensitivity measured in our study is consistent with WHO's minimum target product profile for a non-sputum-based test, with similar sensitivity to sputum Xpert Ultra among a population of predominantly sputum microscopy-positive and HIV-negative individuals. Although the diagnostic performance of saliva Xpert Ultra exceeds WHO's optimal targets for cost ($4) and turnaround time (20 min) for a non-sputum-based test, if Xpert Ultra on saliva could be shown to perform well in high-priority populations for whom sputum collection is difficult, policy makers might be willing to pay more and wait longer for results than specified by WHO's optimal target product profile.

The use of saliva for molecular diagnosis of TB was first described in a convenience sample of 52 adult TB patients in Japan in whom testing of saliva using a lab-developed, nested PCR assay targeting the 65-kDa mycobacterial antigen detected TB in 98% of those with confirmed TB (14). In a more recent study of 44 sputum smear-positive and culture-positive TB patients, including 35 in South Africa and 9 in South Korea, Shenai et al. reported that saliva tested with Xpert Ultra had a very low sensitivity of 39% for active TB disease relative to a reference standard of sputum liquid mycobacterial culture (15). Sputum mycobacterial load was similarly high (100% smear positive in the Shenai et al. study versus 87% in our study), so these differences in diagnostic performance might be attributable to differences in sample collection, specimen processing, or the molecular assays used. For example, participants in the Shenai et al. study were instructed to rinse their mouths prior to specimen collection and to provide saliva using a chewable cotton swab, different from the protocol used in our study. Second, Shenai et al. used a 2:1 sample reagent (SR)-to-saliva dilution ratio following the manufacturer's guidelines for sputum processing, while we used a 1:1 dilution ratio as recommended for cerebrospinal fluid, another extrapulmonary specimen without a mucoid matrix (16). Finally, we used the Xpert Ultra cartridge, which has a 10-fold-lower limit of detection threshold than the earlier-generation GeneXpert MTB/RIF cartridge used in the Shenai et al. study (11). To our knowledge, we are among the first to report the performance of Xpert Ultra on saliva.

Previous studies have examined the sensitivity of a variety of oral specimens for diagnosis of TB. A study using oral swab analysis to detect *M. tuberculosis* in pulmonary TB patients in Uganda showed a sensitivity of 88% on day 1 and 94% on day 2 for single swabs tested with IS*6110* PCR relative to a reference standard of sputum Xpert Ultra (17). A recent study from the United States was among the first to show that saliva is a viable and accurate specimen for diagnosis of SARS-CoV-2 and is more sensitive and less variable than nasopharyngeal swab specimens (9). Another study carried out in Thailand using saliva for diagnosis of SARS-CoV-2 found similar results, with saliva providing a sensitivity of 84% and a specificity of 99% (18). Collectively, these studies suggest that saliva is a very promising novel specimen for diagnosis of respiratory tract infections.

There were a few limitations to our study. First, our study population included an exceptionally high proportion of sputum smear-positive individuals. In addition, because our primary study objective was to evaluate feasibility and preliminary sensitivity, we did not include children or patients with nonproductive cough, two ideal target populations for salivary testing who are likely to have more paucibacillary disease. Diagnostic sensitivity is likely to be lower in paucibacillary populations, as suggested by the lower sensitivity that we observed among sputum smear-negative individuals and among persons living with HIV. However, in the current study, we found that even though saliva is more paucibacillary than sputum, as assessed by Xpert Ultra's semiquantitative measurement of mycobacterial load, diagnostic sensitivity was similar between the two specimen types, likely because of the extremely low threshold of detection and high analytic sensitivity of the Xpert Ultra assay (11).

Second, to conserve costs in this preliminary study, we enrolled only individuals with confirmed TB, as documented by positive sputum Xpert Ultra results. Although case-only studies are more cost-efficient, they have the limitation of potentially inflating sensitivity by excluding patients who would have been diagnosed with TB by a more sensitive inclusion criterion, such as sputum mycobacterial culture or clinical evaluation (19). However, we found that only 2 of the 153 patients screened for this study were likely to be sputum Xpert Ultra negative and culture positive, making the selection bias too small to meaningfully influence our sensitivity estimates. In addition, by not enrolling non-TB patients to serve as controls, we were unable to estimate diagnostic specificity for salivary Xpert Ultra testing. A recent systematic review found that the previous generation Xpert MTB/RIF and/or Xpert Ultra assays have a high specificity on a variety of body fluid types, including pleural fluid, peritoneal fluid, pericardial fluid, lymph node aspirates, bone or joint aspirates, urine, and blood, although no studies of saliva Xpert were identified (20). Studies of saliva Xpert Ultra specificity are now needed because the oral cavity is in direct contact with environmental air, increasing the risk of false-positive results in high-transmission environments, including health care facilities. Third, our sample size was small, with relatively few persons living with HIV and few sputum smear-negative patients, which limited our ability to develop precise accuracy estimates for these and other subgroups.

Fourth, we did not evaluate the optimal SR-to-saliva ratio. Unlike sputum, saliva may not require processing with a mucolytic. Therefore, reducing the SR-to-saliva ratio from the standard 2:1 ratio to a 1:1 ratio, as we did, could improve analytic sensitivity by minimizing dilution of target DNA. However, SR also plays an important role in sterilizing mycobacteria, and reducing the final concentration of SR could create a biosafety risk if processing occurs outside a level II biosafety cabinet, as may commonly occur in low- and middle-income settings. Both the risk of aerosolizing *M. tuberculosis* from samples during Xpert MTB/RIF processing and the effectiveness of SR in sterilizing samples at ratios ≥2:1 have been well documented in sputum (21 to 23), but not in saliva. Future studies should assess the aerosolization risk in saliva and assess the sterilizing effectiveness of SR in saliva at different dilutions. Last, because saliva was collected after sputum, residual bacilli left in the mouth after coughing may have exaggerated the diagnostic yield of saliva collected without cough. Alternatively, expectoration may have depleted the mouth of *M. tuberculosis* bacilli, reducing sensitivity. Future studies should compare the diagnostic yield of Xpert Ultra with and without coughing before saliva collection.

In conclusion, saliva appears to be a feasible specimen for TB diagnosis using Xpert Ultra, with a similar diagnostic sensitivity to sputum Xpert Ultra among a population of predominantly HIV-negative and sputum smear-positive individuals. Thus, saliva Xpert Ultra appears to be a very promising nonsputum diagnostic test for active TB in high-burden settings. Future studies should examine sensitivity and specificity in larger and broader populations. These should include (i) those most likely to benefit from this test, such as children and individuals who are unable to expectorate sputum; (ii) those in whom diagnostic sensitivity is uncertain, including populations with paucibacillary disease; and (iii) symptomatic individuals without TB, including persons living with HIV, to estimate diagnostic specificity. Direct comparisons of saliva to other oral sampling methods, including swabs, would also be useful. Finally, studies evaluating the relative impacts of salivary versus sputum testing on infection control proxies and/or outcomes would also be valuable.

## MATERIALS AND METHODS

**Study design and population.** Between June 2018 and May 2019, we carried out a prospective, observational study to determine the performance of Xpert Ultra testing on saliva for diagnosis of active TB. This was a substudy nested within the Mulago Inpatient Noninvasive Diagnosis of Pneumonia-Inflammation Aging, Microbes, and Obstructive Lung Disease (I AM OLD) study. The parent study enrolled consecutive adults (age ≥ 18 years) with cough of any duration for <6 months who were also undergoing TB evaluation (including HIV testing, chest radiography, and sputum examination) as inpatients or outpatients at China-Uganda Friendship Hospital-Naguru in Kampala, Uganda; patients with a

history of TB within the past 2 years, including those receiving treatment for active TB at the time of presentation were excluded. In this substudy, we included consecutive patients with positive sputum Xpert Ultra results at any semiquantitative threshold.

**Measurements and study procedures.** After obtaining written informed consent from participants, a study nurse collected demographic and clinical information using a structured questionnaire and then provided standardized instructions to expectorate sputum into three separate cups "on the spot" (24). Trained study staff examined the first sample using direct auramine O AFB fluorescence microscopy (FM) (25, 26) and sent the remaining sample for mycobacterial culture and species identification on Lowenstein-Jensen (LJ) solid medium and in mycobacterial growth indicator tube (MGIT) liquid medium, the accepted microbiologic reference standard assays for TB. Staff examined the second sample using direct FM and performed Xpert Ultra testing on the remainder (27). Finally, the staff sent a third sputum sample for mycobacterial culture on solid media and liquid culture. All cultures were performed at the Makerere University Mycobacteriology Laboratory, and the staff performing the cultures were not provided with clinical information about the study participants.

After sputum collection, participants were asked not to eat or drink before saliva collection. At least 2 h after sputum collection, they were then instructed to deposit ≥1 mL of saliva in a sterile 50-mL conical specimen cup, taking care not to intentionally cough before saliva collection. Saliva was immediately transferred to the laboratory for Xpert Ultra testing. Saliva specimens were processed for Xpert Ultra using a sample reagent to saliva volume ratio of 1:1, and all other steps followed the manufacturer's recommendations for extrapulmonary body fluid specimens (27). Sputum was collected prior to TB treatment initiation, and saliva was collected prior to or within 2 h of TB treatment initiation. Finally, all participants without a prior known HIV diagnosis were offered HIV testing and counseling; for those found to be living with HIV, a CD4-positive ($CD4^+$) T-cell count was performed at the Makerere University-Johns Hopkins University Research Collaboration (MU-JHU) laboratory.

**Statistical analysis.** We examined baseline characteristics using proportions for categorical variables and medians for continuous variables. We calculated sensitivity for Xpert Ultra results on saliva and sputum in reference to a primary reference standard of composite sputum mycobacterial culture, described as follows: those with ≥1 sputum sample culture positive were defined as *Mycobacterium tuberculosis* positive, those with two negative cultures were defined as negative, and all others were defined as indeterminate. As a secondary analysis, we also calculated sensitivity relative to a sputum molecular reference standard, for which we defined any positive sputum Xpert Ultra result greater than trace as *M. tuberculosis* positive. We excluded patients with missing saliva samples and those with indeterminate culture results from the analysis. We estimated precision using exact binomial 95% confidence intervals. We explored comparisons of diagnostic accuracy results (sensitivity differences with 95% CI) and semiquantitative results (Fisher's exact test) for saliva Xpert Ultra by sputum smear microscopy and HIV status. We estimated that a sample size of 84 patients would enable us to determine if the sensitivity of saliva Xpert Ultra were ≥75% with a precision of ±10%, allowing for up 10% indeterminate results due to missing or contaminated sputum culture results. We used Stata 14.0 (Stata Corporation, College Station, TX) for all statistical analyses.

**Human subject protection.** The study protocol was reviewed and approved by Yale University and the University of California San Francisco Institutional Review Boards, the Makerere University School of Medicine Research Ethics Committee, the Mulago Hospital Institutional Review Board, and the Uganda National Council for Science and Technology.

**Data sharing.** A comprehensive, deidentified data set containing individual-level data is publicly available for download (28).

## ACKNOWLEDGMENTS

We acknowledge the patients who participated in the study and the administration and TB clinic staff of the China-Uganda Friendship Hospital where the study took place.

This work was supported in part by NIH D43 TW009607 (J.L.D.), the Pulmonary Complications of AIDS Research Training (PART) program, and NIH K24 HL087713 (L.H.) and R01 HL128156 (L.H.).

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
