## [Reviewer comments · Microbiology Spectrum]

Microbiology Spectrum

Feasibility and Sensitivity of Saliva GeneXpert MTB/RIF Ultra for Tuberculosis Diagnosis in Adults in Uganda

Patrick Byanyima, Sylvia Kaswabuli, Emmanuel Musisi, Catherine Nabakiibi, Josephine Zawedde, Ingvar Sanyu, Abdul Ssesolo, Alfred Andama, William Worodria, Laurence Huang, and J. Davis

Corresponding Author(s): J. Davis, Yale School of Public Health

Review Timeline:

Submission Date:	March 25, 2022
Editorial Decision:	June 8, 2022
Revision Received:	August 17, 2022
Accepted:	August 30, 2022

Editor: Rita Oladele

Reviewer(s): Disclosure of reviewer identity is with reference to reviewer comments included in decision letter(s). The following individuals involved in review of your submission have agreed to reveal their identity: Rachel C. Wood (Reviewer #1)

Transaction Report:

DOI: <https://doi.org/10.1128/spectrum.00860-22>

June 8, 2022

Dr. J. Lucian Davis
Yale School of Public Health
New Haven, CT

Re: Spectrum00860-22 (Feasibility and Sensitivity of Saliva GeneXpert MTB/RIF Ultra for Tuberculosis Diagnosis in Adults in Uganda)

Dear Dr. J. Lucian Davis:

Link Not Available

Sincerely,

Rita Oladele

Journals Department
Comments:

Abstract appears misleading in its characterization of the study population, which seems to imply that all persons being evaluated for TB were included in the study population. when in fact, it was only sputum Xpert positive that were included in this sub-study. By not including those who were sputum Xpert negative (but could have been sputum culture positive), the sensitivity is likely higher than would have been observed had everyone in the parent study been included. The reviewers suggest a revision of the manuscript to describe a feasibility study and minimize any discussion of sensitivity. I agree with them

Additionally, there is have biosafety concerns regarding the lower concentration of Cepheid Sample Reagent used to treat the saliva samples. Sample Reagent does inactivate bacilli in sputum when used at the manufacturer's recommendations. To my knowledge, inactivation rates of lower concentrations of Sample Reagent have not been evaluated in saliva. While this concentration may still provide adequate inactivation, this should be evaluated, or samples should be handled in a biosafety cabinet in order to ensure the safety of lab personnel.

Reviewer #1 (Comments for the Author):

This manuscript reports the important finding that saliva samples analyzed by GeneXpert Ultra may serve as a non-sputum alternative for TB diagnosis. Identifying viable non-sputum samples is critical to the fight against TB, and it can greatly benefit the patients and healthcare workers.

I would consider the study presented in this manuscript to be a successful proof-of-concept study, where the population and methods gave the saliva samples the best chance to perform well relative to sputum testing. It is certainly key step in establishing saliva as a diagnostic sample, though additional research will be necessary before saliva is more widely collected and used.

Specifically, the following points should be addressed:

Line 20: Replace "of" with "for".

Line 22: Delete "for example".

Line 24: Add specific sensitivity data for the study referenced (Reference 6).

Lines 28-29: A citation is needed here.

Line 56: "Speciation" is spelled incorrectly.

Lines 62-64: By collecting saliva after sputum, it is possible that the bacilli from the sputum remained in the mouth. I'd like this addressed as a caveat/limitation of this study and for follow-up studies to collect saliva before sputum (or at least compare before vs after).

Line 118: Saliva Xpert sensitivity relative to sputum Xpert, and sputum Xpert sensitivity relative to the combined sputum culture reference standard should also be calculated and included here.

Line 153: Clarify what "sensitivity of 98%" refers to. Is it the PCR assay or the diagnostic sensitivity of saliva in reference 15? If it is the diagnostic sensitivity of saliva, what is it relative to? Sputum?

Line 155: When you state "very low sensitivity of 39% for TB testing [16]" what sample/test is this relative to? And were the samples tested in reference 16 analyzed by GeneXpert or manual PCR?

Lines 170-173: I recommend referencing a more recent oral swab study, such as Luabeya et al 2018, or Wood/Andama et al, 2021.

Line 192: Reference 23 doesn't test saliva, so it is not relevant. It should be removed.

Lines 180-194: To the paragraph on limitations, add that saliva was collected after sputum, which may have biased the results. It could go either way; expectoration could have depleted the available bacilli in the mouth or coughing up sputum could have deposited additional bacilli in the mouth.

Line 197: Saliva Xpert sensitivity relative to sputum Xpert needs to be provided here, or earlier in the manuscript.

Reviewer #3 (Comments for the Author):

Lines 23-26. The study described in this manuscript does not address the objective described in the first sentence of the abstract. The study does not assess the performance of the test with saliva samples from persons being evaluated for TB. Rather, it assesses the performance of test with saliva samples from persons being evaluated for TB who have produce sputum specimens that are Xpert positive. These are very different populations of test subjects.

Lines 28-29 and lines 155-157. The sensitivity compared to culture was 93.6% (73/78), not 90%.

Lines 32 to 34 (also lines 162-164). The sensitivity was 46% higher (not lower) in smear-positive vs smear-negative.

A potential confounding aspect of the study design is that the saliva samples were collected after the sputum samples were collected. It is not obvious that waiting 2 hours is sufficient time to avoid potential 'contamination' of the saliva samples with bacteria introduced into the mouth during sputum collection.

The authors should discuss how the data from this somewhat unusual population of test subjects can be extrapolated to the typical population of persons being evaluated for TB. That is, in the overall pool of recruited test subjects, more than half were bacteriologically confirmed TB patients and almost half were smear positive. This pool of test subjects was further restricted to Xpert-positive test subjects, 87% of whom were smear positive.

Lines 177-178, The 'rigorously defined reference standard' should be clearly stated.

Line 195. 44 smear-negative samples? Line 198 suggests it should be 44 smear-positive samples.

GeneXpert refers to the instrument. The test is the Xpert MTB/RIF Ultra test. The authors should take care to refer to the test in such a way as to avoid confusion between the Xpert MTB/RIF test and the Xpert MTB/RIF Ultra test.

Reviewer #4 (Public repository details (Required)):

Authors indicate in Lines 133-134 that a dataset will be provided prior to publication.

Reviewer #4 (Comments for the Author):

Introduction

Thank you for the opportunity to review this manuscript. The authors have provided evidence that saliva may be a suitable specimen for the diagnosis of TB, while acknowledging that future studies are needed in order to address lingering questions. The authors collected saliva in a population of sputum Xpert positive patients and showed that there were similar rates of detection of tuberculosis in saliva Xpert Ultra as compared to sputum smear microscopy using sputum culture as the reference standard. While a population of sputum Xpert positive patients is convenient, it does raise the concern that such a population is more likely to test positive with saliva Xpert Ultra as well as sputum culture, thus inflating the sensitivity higher than would be observed in a sputum Xpert negative population. Saliva samples were treated at a 1:1 ratio with Cepheid's Sample Reagent. There are potential biosafety considerations using this lower concentration of Sample Reagent that the authors did not address in this study. The authors are encouraged to comment on biosafety concerns as well as temper their conclusions given the selective population in this study.

Major Comments

- 1. The authors chose to treat the saliva samples with Cepheid Sample Reagent at a 1:1 ratio, which is used for some other non-pulmonary samples such as cerebrospinal fluid. In contrast, sputum is treated at a 2:1 Sample Reagent to sputum ratio. As reported, other groups, such as the South Africa/South Korea study (1) had treated saliva at a 2:1 ratio. The purpose of the sample reagent is two-fold, to liquefy viscous samples, as well as to render bacilli unviable and thus lowering the biohazard risk. As reported in Helb et al. (2) killing assays in spiked sputum were utilizing two volumes of Sample Reagent per volume of sputum. That group showed that at that concentration, after 15 minutes, viability was reduced by at least 8-logs in sputum. This is an important consideration in the safety of laboratory staff to ensure that they are not unnecessarily exposed to viable bacilli. To the reviewer's knowledge, no such killing assays have been conducted for saliva samples at a lower Sample Reagent concentration. If samples are handled within a biosafety cabinet at all times, concerns regarding a lower Sample Reagent concentration are alleviated; however, if treated samples are opened on an open bench, lab workers could potentially be exposed to viable bacilli. The authors are encouraged to specify that samples should be handled within a biosafety cabinet at all times or conduct killing assays to show that lower concentrations of Sample Reagent still result in an adequate reduction in bacilli viability.**
- 2. In the Abstract, the authors state, "The objective... was to determine the performance of GeneXpert MTB/RIF Ultra (Xpert) testing on saliva for active tuberculosis (TB) disease among consecutive adults undergoing diagnostic evaluation." This statement is misleading. While the parent study did enroll consecutive adults undergoing TB evaluation, in the sub-study, saliva samples were only collected among adults that had already tested positive for tuberculosis using sputum Xpert. This has important implications in the interpretation of the reported sensitivity. As is noted in the Discussion regarding Wood et al. (3), case-control study designs are prone to inflate diagnostic accuracy. The authors of this study do not acknowledge this same limitation in their study design, which is effectively a "case-only" study population. The authors do note the limitation of not being able to determine specificity, but do not note the limitation of inflated sensitivity. The authors are encouraged to acknowledge this major limitation throughout the manuscript.**
- 3. Saliva was collected "at least two hours after sputum collection." Is this a sufficient period of waiting? Why was saliva not collected before sputum? It could be reasonably assumed that the process of expectorating sputum would leave some bacilli in the mouth that could then later be released in saliva. Had the order of collection been reversed (saliva first and then sputum), one might expect detection in saliva Xpert to be lower. The authors are strongly encouraged to discuss the rationale for this methodology, and discuss whether future studies should explore sample collection order.**

1) Shenai S, Amisano D, Ronacher K, et al. Exploring alternative biomaterials for diagnosis of pulmonary tuberculosis in HIV-negative patients by use of the GeneXpert MTB/RIF assay. *J Clin Microbiol.* 2013;51(12):4161-4166. doi:10.1128/JCM.01743-13

2) Helb D, Jones M, Story E, et al. Rapid Detection of *Mycobacterium tuberculosis* and Rifampin Resistance by Use of On-Demand, Near-Patient Technology. *J Clin Microbiol.* 2010;48(1):229-237. doi:10.1128/JCM.01463-09

3) Wood RC, Luabeya AK, Weigel KM, et al. Detection of *Mycobacterium tuberculosis* DNA on the oral mucosa of tuberculosis patients. *Sci Rep.* 2015 Mar 2;5:8668. doi:10.1038/srep08668

Minor Comments

- 1. Line 102: The authors are encouraged to explain the instructions provided to patients for collecting saliva samples in greater detail to ensure that other groups are able to recreate sampling as closely as possible.**
- 2. Table 2: The authors are encouraged to include a negative row and column to show those individuals positive only in saliva Xpert and those positive only in sputum Xpert.**

Staff Comments:

Preparing Revision Guidelines

For complete guidelines on revision requirements, please see the journal Submission and Review Process requirements at <https://journals.asm.org/journal/Spectrum/submission-review-process>. Submissions of a paper that does not conform to Microbiology Spectrum guidelines will delay acceptance of your manuscript. "

Please return the manuscript within 60 days; if you cannot complete the modification within this time period, please contact me. If you do not wish to modify the manuscript and prefer to submit it to another journal, please notify me of your decision immediately so that the manuscript may be formally withdrawn from consideration by Microbiology Spectrum.

If your manuscript is accepted for publication, you will be contacted separately about payment when the proofs are issued; please follow the instructions in that e-mail. Arrangements for payment must be made before your article is published. For a complete list of Publication Fees, including supplemental material costs, please visit our website.

**Feasibility and Sensitivity of Saliva GeneXpert MTB/RIF Ultra for Tuberculosis Diagnosis**
**in Adults in Uganda**

Patrick Byanyima¹, Sylvia Kaswabuli¹, Emmanuel Musisi², Catherine Nabakiibi¹, Josephine
Zawedde¹, Ingvar Sanyu¹, Abdul Sessolo¹, Alfred Andama^{1,5}, William Worodria^{1,5}, Laurence
Huang^{3,4}, J. Lucian Davis^{6,7}

1. Infectious Diseases Research Collaboration, Kampala, Uganda
2. Division of Infection and Global Health, School of Medicine, University of St. Andrews, UK
3. Division of Pulmonary and Critical Care Medicine, University of California San Francisco,
San Francisco, California, USA
4. Division of HIV, Infectious Diseases, and Global Medicine, University of California San
Francisco, San Francisco, California, USA
5. Department of Internal Medicine, Makerere University College of Health Sciences, Uganda
6. Department of Epidemiology of Microbial Diseases, Yale School of Public Health, New
Haven, Connecticut, USA
7. Pulmonary, Critical Care, and Sleep Medicine Section, Yale School of Medicine, New
Haven, Connecticut, USA

Running Head: Saliva TB NAAT in Uganda

Word Count (Main Text): 2533 words

**ABSTRACT**

The objective of this prospective, observational study carried out at China-Uganda Friendship
Hospital-Naguru in Kampala, Uganda, was to determine the performance of GeneXpert
MTB/RIF Ultra (Xpert) testing on saliva for active tuberculosis (TB) disease among consecutive
adults undergoing diagnostic evaluation. We calculated sensitivity to determine the diagnostic
performance in comparison to that of the composite reference standard of *Mycobacterium*
*tuberculosis* liquid and solid cultures on two spot sputum specimens. GeneXpert Ultra on saliva
had a sensitivity of 90% (95% confidence interval [CI], 81-96%); this was similar to that of
sputum fluorescence smear microscopy (FM) of 87% (95% CI, 77-94%). Sensitivity was 24%
lower (95% CI for difference 2-48%, p=0.003) among persons living with HIV (71%, 95%CI 44-
90%) than among persons living without HIV (95%, 95%CI 86-99%) and 46% lower (95% CI
for difference 14-77%, p<0.0001) among sputum microscopy positive (96%, 95% CI 87-99%)
than among sputum microscopy negative patients (50%, 95% CI 19-81%). Semi-quantitative
Xpert grade was higher in sputum than in paired saliva samples from the same patient. In
conclusion, saliva specimens appear to be feasible and similarly sensitive to sputum for active
TB diagnosis using molecular testing, suggesting promise as a non-sputum diagnostic test for
active TB in high-burden settings.

Word Count (Abstract): 202 words (Limit 250)

INTRODUCTION

Over the last quarter century, improvements in diagnosis and treatment of people with tuberculosis (TB) have gradually reduced mortality, but large gaps in detection and treatment persist that contribute to substantial ongoing morbidity and mortality [1]. Among several available strategies to facilitate rapid, same-day diagnosis of TB, testing sputum with the GeneXpert MTB/RIF Ultra molecular assay [2, 3] is the most sensitive and most readily available approach. Unfortunately, there are several operational challenges associated with collecting sputum for diagnosis of pulmonary TB. First, coughing during sputum expectoration or sputum induction generates aerosols that may facilitate TB transmission [4]. Second, some individuals are unable to produce sputum, including young children, those with dry cough, and the severely ill/severely debilitated. Given these limitations of sputum for TB diagnosis, in 2014 the World Health Organization (WHO) issued guidelines for developers of a future non-sputum test for active TB diagnosis, including a target product profile suggesting that it should have a minimum diagnostic accuracy similar to sputum GeneXpert MTB/RIF on sputum smear-negative individuals (*i.e.*, sensitivity $\geq 68\%$, specificity $\geq 98\%$) [5].

One alternative sample type with great promise for diagnosis of pulmonary TB is saliva, which is easy to collect, with minimal risk of generating aerosols. Although Stop TB Partnership guidelines discourage collection of salivary sputum samples because they have lower diagnostic yield for acid-fast bacilli (AFB) by microscopy or culture, the diagnostic yield of TB molecular testing appears to be more promising. In a previous study of 1782 smear-negative adults undergoing evaluation for active TB, for example, we found that salivary sputum provided a substantially higher diagnostic yield and sensitivity for culture-positive TB than other sputum

types, implying incremental value to using oral samples at least as a supplement to sputum [6].
Using a different sampling technique, oral swabs, Wood and colleagues showed that oral nylon
swabs repeatedly tested positive for TB via IS6110 polymerase chain reaction testing in 90% of
South African patients with TB confirmed by sputum GeneXpert MTB/RIF testing, suggesting
that TB is present in the oral cavity [7]. A subsequent study of 50 adults with possible TB in
Uganda found similar sensitivity of 88%, albeit with lower specificity. Saliva is also now widely
used for molecular diagnosis of COVID-19, where it has high sensitivity, even among patients
without symptoms [8]. Using saliva as a diagnostic specimen in the COVID-19 context has been
shown to reduce aerosol exposure for health workers and eliminate the need for personal
protective equipment because it is self-collected [9]. This raises the possibility that saliva alone
could be used as a TB diagnostic when paired with next generation and ultra-sensitive molecular
tests (GeneXpert MTB/RIF Ultra). Thus, the aim of this study was to evaluate the feasibility and
sensitivity of GeneXpert MTB/RIF on saliva among symptomatic adult TB confirmed patients.

**MATERIALS AND METHODS**

*Study design & Population.* Between June 2018 and May 2019, we carried out a prospective,
observational study to determine the performance of GeneXpert MTB/RIF Ultra (Xpert) testing
on saliva for diagnosis of active TB. This was a sub-study nested within the Mulago Inpatient
Non-invasive Diagnosis of Pneumonia–Inflammation Aging, Microbes, and Obstructive Lung
Disease (I AM OLD) study. We enrolled consecutive adults (age ≥ 18 years) with cough of any
duration but < 6 months who were also undergoing TB evaluation (including HIV testing, chest
radiography, and sputum examination) as inpatients or outpatients at China-Uganda Friendship
Hospital-Naguru in Kampala, Uganda; patients with a prior history of TB within the past two

47 years and those receiving treatment for active TB at the time of presentation were excluded. In
this sub-study, we included consecutive patients with positive sputum Xpert results at any semi-
quantitative threshold.

***Measurements and Study Procedures:*** After obtaining written informed consent from
participants, a study nurse collected demographic and clinical information using a structured
questionnaire, and then provided standardized instructions to expectorate sputum into three
separate cups “on the spot” [10]. Trained study staff examined the first sample using direct
auramine-O fluorescence microscopy (FM) [11, 12] and sent the remaining sample for
mycobacterial culture and speciaion on Lowenstein-Jensen (LJ) solid media and in
Mycobacterial Growth Indicator Tube (MGIT) liquid media, the accepted microbiologic
reference standard assays for TB. Staff examined the second sample using direct FM and
performed GeneXpert MTB/RIF testing on the remainder [13]. Finally, staff sent a third sputum
sample for mycobacterial culture on solid media and liquid culture.. All cultures were performed
at the Makerere University Mycobacteriology Laboratory, and staff performing the cultures were
not provided with clinical information about the study participants. At least two hours after
sputum collection, the patients were asked to submit at least 1 mL of saliva placed into a sterile
specimen cup for GeneXpert MTB/RIF testing; all participants were instructed not to cough prior
to saliva collection. Saliva specimens were processed for GeneXpert MTB/RIF using a sample
reagent to saliva volume ratio of 1:1, and all other steps followed the manufacturer’s
recommendations for extra-pulmonary body fluid specimens [13]. Sputum was collected prior to
TB treatment initiation, and saliva was collected prior to or within two hours of TB treatment
initiation. Finally, all participants without a prior known HIV diagnosis were offered HIV testing

and counseling, and for those found to be living with HIV, a CD4+ T-cell count was performed
at the Makerere University–Johns Hopkins University Research Collaboration (MU-JHU)
laboratory.

***Statistical Analysis:*** We examined baseline characteristics using proportions for categorical
variables, and medians for continuous variables. We calculated sensitivity for GeneXpert
MTB/RIF results on saliva and on sputum in reference to a composite reference standard
described as follows: those with ≥ 1 sputum sample culture-positive were defined as
*Mycobacterium tuberculosis (Mtb)* positive, those with two negative cultures were defined as
negative, and all others were defined as indeterminate. We estimated precision using exact
binomial 95% confidence intervals. We explored comparisons of diagnostic accuracy results
(sensitivity differences with 95% CI) and semi-quantitative results (Fisher’s exact test) for saliva
GeneXpert by sputum smear microscopy and HIV status. We estimated that a sample size of 84
patients would enable us to determine if the sensitivity of saliva GeneXpert MTB/RIF was 75%
or higher with a precision of $\pm 10\%$, allowing for up to 10% indeterminate results due to missing or
contaminated sputum culture results. We used STATA 14.0 (Stata Corporation, College Station,
TX) for all statistical analyses.

***Human subjects protection.*** The study protocol was reviewed and approved by the Yale
University and the University of California San Francisco Institutional Review Boards, the
Makerere University School of Medicine Research Ethics Committee, the Mulago Hospital
Institutional Review Board, and the Uganda National Council for Science and Technology.

**Data sharing.** A comprehensive, de-identified dataset containing individual-level data will be
made available prior to publication.

**RESULTS**

**Study Population.** Among 153 participants enrolled into the parent study between June 2018 and
June 2019, 40 were GeneXpert MTB/RIF negative; 15 did not have sputum GeneXpert
MTB/RIF performed; 16 did not provide a saliva specimen; and one participant had an
indeterminate culture result, (Figure 1) leaving 81 participants for inclusion in the analysis.
There were no adverse events during specimen collection. Median age of participants was 30
102 years (interquartile range 24-38), 50 (62%) were men. 18 (22%) were persons living with HIV,
with median CD4 cell count 90 cells/uL (interquartile range 49-234), and only seven of the 18
(39%) were taking antiretroviral therapy at enrolment. 17 (21%) had ever smoked ≥ 100
cigarettes in their entire life and 60 (74%) had a history of alcohol use. 17 (21%) had a cough
greater than two weeks, while 71 (88%) reported subjective fever within the past seven days. 75
(93%) reported weight loss, including 46 (57%) with weight loss ≥ 5 kg. 14 (17%) reported no
ambulatory limitation; 47 (58%) were mildly limited with ambulation, and 20 (25%) were
severely affected but not bedbound. 67 patients (83%) were AFB smear-positive, including 30
(37%) with an AFB microscopy smear grade of 3+, 18 (22%) with a grade of 2+, nine (11%) 1+,
and 10 (12%) had 1-9 AFB seen per 100 high-powered fields. 13 (16%) were AFB smear-
negative and one (1%) was missing an AFB smear microscopy result (Table 1).

**Diagnostic Performance.** Seventy-eight patients were confirmed *Mtb* culture-positive on liquid
and/or solid media, while three were *Mtb* culture-negative. Seventy-three of the 78 patients with

culture-confirmed TB were salivary GeneXpert MTB/RIF positive, giving an overall sensitivity
of GeneXpert MTB/RIF on saliva of 90% (95% Confidence Interval (CI) 81-96%). This
sensitivity was similar to that of sputum smear microscopy, which had a sensitivity of 87% (95%
CI 77-94%) in reference to the combined culture reference standard. Among the three *Mtb*
culture-negative patients, two were salivary GeneXpert positive. Sensitivity was 24% lower
(95% CI for difference 2-48%, $p=0.003$) among persons living with HIV (71%, 95% CI 44-90%)
than among persons living without HIV (95%, 95% CI 86-99%), and 46% lower (95% CI for
difference 14-77%, $p<0.0001$) among sputum microscopy positive (96%, 95% CI 87-99%) than
among sputum microscopy negative patients (50%, 95% CI 19-81%).

We also compared the semi-quantitative results of bacilli by GeneXpert for both saliva and
sputum, as shown in Table 2. Overall, the semi-quantitative GeneXpert grade was higher in
sputum samples than in paired saliva samples collected from the same patient: 56 of 72 (78%) of
the sputum samples of either medium ($n=22$) or high ($n=34$) semi-quantitative grade, whereas
only 14 of 72 (19%) of the saliva samples were of either medium ($n=10$) or high ($n=4$) grade,
indicating that the mycobacterial load in the saliva specimens was low overall. There was no
difference in semi-quantitative results by smear microscopy result ($p=0.52$) or by HIV status
($p=0.39$).

**DISCUSSION**

In a prospective, observational study of consecutive sputum GeneXpert-positive TB patients in a
high-burden setting, we showed that diagnosis of TB using GeneXpert Ultra on saliva is feasible
and had a high sensitivity relative to a rigorously defined reference standard. This finding has

significant implications for the diagnosis of TB and potentially also for TB infection control.
Using sputum specimens for TB diagnosis poses a number of challenges, since some individuals
such as those with non-productive cough and young children find expectoration challenging, and
the associated generation of sputum aerosols poses an infection control risk for health care
workers and nearby patients [14]. The development of novel testing strategies that employ non-
sputum samples for TB has been identified as a priority by the WHO, and the sensitivity
measured in our study is consistent with WHO's minimum target-product profile for a non-
sputum-based test, with similar sensitivity to sputum GeneXpert among a population of
predominantly sputum microscopy-positive and HIV-negative individuals. Although our
alternative strategy of salivary GeneXpert exceeds WHO's optimal targets for cost (\$4) and turn-
around time (20 minutes) for a non-sputum-based test, if GeneXpert on saliva were shown to
perform well in populations for whom sputum collection is less feasible for the reasons described
above, the willingness to pay for and wait for results might be higher.

The use of saliva for molecular diagnosis of TB was first described in a convenience sample of
52 adult TB patients in Japan who were evaluated using a lab-developed, nested PCR assay that
was shown to have a sensitivity of 98% [15]. A more recent study of 44 sputum smear- and
culture-positive TB patients, including 35 in South Africa and 9 in South Korea, reported on
saliva as having a very low sensitivity of 39% for TB testing [16]. Sputum mycobacterial load
was similarly high (100% smear-positive in the South Africa/South Korea study vs. 87% in our
study), so these differences in diagnostic performance might be attributable to differences in
either sample collection or specimen processing. For example, participants were instructed to
rinse their mouths prior to specimen collection in the South Africa/South Korea study but not in

our study. Second, the South Africa/South Korea study diluted one part of the sample in two
parts of sample reagent as recommended by the manufacturer for sputum, while we used a 1:1
dilution ratio as recommended for cerebrospinal fluid, another extra-pulmonary specimen
without a mucoid matrix [17]. Finally, we used the GeneXpert MTB/RIF Ultra cartridge, which
has ten-fold better analytic sensitivity than the earlier generation GeneXpert MTB/RIF cartridge.
To our knowledge, we are among the first to report the performance of GeneXpert MTB/RIF
Ultra on saliva.

Previous studies have examined the sensitivity of a variety of oral specimens for diagnosis of
TB. We previously showed that oropharyngeal wash specimens paired with a lab-developed PCR
assay had a high sensitivity for TB diagnosis in reference to sputum mycobacterial culture on
previously frozen and thawed sputum, but a subsequent study failed to confirm these results [18,
19]. A study of *Mtb* PCR on buccal swabs of South African TB patients and US controls showed
high sensitivity (90%) and specificity (100%), although the case-control design may have
inflated diagnostic accuracy [7]. A recent study from the US was among the first to show that
saliva is a viable and accurate specimen for diagnosis of SARS-CoV2, and more sensitive and
less variable than nasopharyngeal swab specimens [20]. Another study carried out in Thailand
using saliva for diagnosis of SARS-CoV2 showed similar results, with saliva providing a
sensitivity of 84% and a specificity of 99% [21]. Collectively, these studies suggest that saliva is
a very promising novel specimen for diagnosis of respiratory tract infections.

There were a few limitations to our study. First, because the primary study objective was to
evaluate feasibility and preliminary sensitivity, we did not include patients with non-productive

cough or children, two ideal target populations for salivary testing. If, as seems plausible, these
populations have more paucibacillary disease, diagnostic sensitivity could be lower in these
populations, as suggested by the lower sensitivity observed among sputum smear-negative
individuals and persons living with HIV. However, in the current study, we found that even
though saliva is more paucibacillary than sputum as assessed by GeneXpert's semi-quantitative
measurement of mycobacterial load, diagnostic sensitivity was similar between the two specimen
types, likely because of the extremely low threshold of analytic sensitivity of the GeneXpert
Ultra assay [22]. Secondly, to conserve costs in this preliminary study, we did not enroll non-TB
patients to serve as controls, a choice that prevented us from estimating diagnostic specificity.
However, a recent systematic review found that both GeneXpert MTB/RIF assays have a high
specificity on a variety of body fluid types [23]. Thirdly, our sample size was small, especially
for persons living with HIV and for sputum smear-negative patients, which limited our ability to
develop precise accuracy estimates for these and other subgroups.

In conclusion, saliva appears to be a feasible specimen for TB diagnosis using GeneXpert Ultra,
with a similar diagnostic sensitivity to sputum GeneXpert Ultra, at least among HIV-negative
and sputum smear-positive individuals. and thus appears to be a very promising alternative non-
sputum diagnostic test for active TB in high-burden settings. Future studies should examine
sensitivity in populations who are most likely to benefit from this test, including individuals who
are unable to expectorate sputum, children, and individuals from populations with a broad
spectrum of mycobacterial load and disease severity, and symptomatic individuals without TB,
including persons living with HIV. Direct comparisons of saliva to other oral sampling methods,

including swabs, would also be useful. Finally, studies evaluating the relative impacts of salivary
versus sputum testing on infection control proxies and/or on outcomes would also be valuable.

**ACKNOWLEDGMENTS**

The authors would like to acknowledge the patients who participated in the study and the
administration and TB clinic staff of the China-Uganda Friendship Hospital where the study took
place.

**FUNDING**

This work was supported in part by NIH D43 TW009607 (JLD), the Pulmonary Complications
of AIDS Research Training (PART) program; and by NIH K24 HL087713 (LH) and R01
HL128156 (LH).

REFERENCES

1. WHO, *Global tuberculosis report 2018*. 2018: Geneva.
2. Horne, D.J., et al., *Xpert MTB/RIF and Xpert MTB/RIF Ultra for pulmonary tuberculosis and rifampicin resistance in adults*. Cochrane Database of Systematic Reviews, 2019(6).
3. Di Tanna, G.L., et al., *Effect of Xpert MTB/RIF on clinical outcomes in routine care settings: individual patient data meta-analysis*. The Lancet Global Health, 2019. 7(2): p. e191-e199.
4. Jones-López, E.C., et al., *Cough aerosols of Mycobacterium tuberculosis predict new infection. A household contact study*. American journal of respiratory and critical care medicine, 2013. 187(9): p. 1007-1015.
5. Profiles, W.H.-P.T.P., *for New Tuberculosis Diagnostics: Report of a Consensus Meeting*. Geneva: World Health Organisation, 2014.
6. Meyer, A.J., et al., *Sputum quality and diagnostic performance of GeneXpert MTB/RIF among smear-negative adults with presumed tuberculosis in Uganda*. PLoS One, 2017. 12(7).
7. Wood, R.C., et al., *Detection of Mycobacterium tuberculosis DNA on the oral mucosa of tuberculosis patients*. Scientific reports, 2015. 5: p. 8668-8668.
8. Wyllie, A.L., et al., *Saliva or nasopharyngeal swab specimens for detection of SARS-CoV-2*. New England Journal of Medicine, 2020. 383(13): p. 1283-1286.
9. Johnson, A.J., et al., *Saliva Testing Is Accurate for Early-Stage and Presymptomatic COVID-19*. 2021. 9(1): p. e0008621.
10. Khan, M.S., et al., *Improvement of tuberculosis case detection and reduction of discrepancies between men and women by simple sputum-submission instructions: a pragmatic randomised controlled trial*. The Lancet, 2007. 369(9577): p. 1955-1960.
11. Kent, P.T., *Public health mycobacteriology: a guide for the level III laboratory*. 1985: US Department of Health and Human Services, Public Health Service, Centers ...
12. McCarter, Y.S. and A. Robinson, *Detection of acid-fast bacilli in concentrated primary specimen smears stained with rhodamine-auramine at room temperature and at 37 degrees C*. Journal of clinical microbiology, 1994. 32(10): p. 2487-2489.

13. Hillemann, D., et al., *Rapid Molecular Detection of Extrapulmonary Tuberculosis by the Automated GeneXpert MTB/RIF System*. Journal of Clinical Microbiology, 2011. **49**(4): p. 1202.
14. Namuganga, A.R., et al., *Suitability of saliva for Tuberculosis diagnosis: comparing with serum*. BMC infectious diseases, 2017. **17**(1): p. 600-600.
15. Eguchi, J., et al., *PCR method is essential for detecting Mycobacterium tuberculosis in oral cavity samples*. Oral microbiology and immunology, 2003. **18**(3): p. 156-159.
16. Shenai, S., et al., *Exploring Alternative Biomaterials for Diagnosis of Pulmonary Tuberculosis in HIV-Negative Patients by Use of the GeneXpert MTB/RIF Assay*. Journal of Clinical Microbiology, 2013. **51**(12): p. 4161-4166.
17. Organization, W.H., *Xpert MTB/RIF implementation manual: technical and operational 'how-to'; practical considerations*. 2014, World Health Organization.
18. Davis, J.L., et al., *Polymerase chain reaction of secA1 on sputum or oral wash samples for the diagnosis of pulmonary tuberculosis*. Clinical infectious diseases, 2009. **48**(6): p. 725-732.
19. Davis, J.L., et al., *Nucleic acid amplification tests for diagnosis of smear-negative TB in a high HIV-prevalence setting: a prospective cohort study*. PLoS One, 2011. **6**(1): p. e16321.
20. Wyllie, A.L., et al., *Saliva or Nasopharyngeal Swab Specimens for Detection of SARS-CoV-2*. New England Journal of Medicine, 2020. **383**(13): p. 1283-1286.
21. Pasomsab, E., et al., *Saliva sample as a non-invasive specimen for the diagnosis of coronavirus disease 2019: a cross-sectional study*. Clinical Microbiology and Infection, 2021. **27**(2): p. 285.e1-285.e4.
22. Chakravorty, S., et al., *The New Xpert MTB/RIF Ultra: Improving Detection of Mycobacterium tuberculosis and Resistance to Rifampin in an Assay Suitable for Point-of-Care Testing*. MBio, 2017. **8**(4).
23. Kohli, M., et al., *Xpert MTB/RIF Ultra and Xpert MTB/RIF assays for extrapulmonary tuberculosis and rifampicin resistance in adults*. Cochrane Database Syst Rev, 2021. **1**(1): p. Cd012768.

FIGURES AND TABLES

FIGURE 1. Flow diagram showing study enrollment and TB testing results.

Legend: Patients missing index test results due to a missing saliva sample and patients with missing reference standard results due to indeterminate culture results were excluded from analysis.

TABLE 1. Demographic and clinical characteristics.

Characteristic (n=81)	n (%)*
Age, median years (Q1-Q3)	30(24-38)
Men	50 (62)
Inpatients	8 (9)
Persons living with HIV	18(22)
CD4+ T-cell count, median cells/mL (Q1-Q3)	90 (49-234)
Taking antiretroviral therapy at enrollment	7 (39)
Smoking history	17 (21)
Alcohol use	60 (74)
Cough of any duration	81 (100)
Cough for \geq 14 days	17 (21)
Fever within the past 7 days	71 (88)
Weight loss	75 (93)
Difficulty in breathing within the past 7 days	46 (47)
Clinical status	
Ambulatory, unaffected	14 (17)
Ambulatory, mildly affected	47 (58)
Bedbound, moderately affected	12(15)
Bedbound, severely affected	8 (10)
Sputum AFB microscopy grade [†]	
3+	30 (37)
2+	18(22)
1+	9 (11)
1-9 per 100 hpf	10 (12)
Negative	13 (16)

Abbreviations: AFB, acid-fast bacilli; hpf, high-powered fields; Q1, 25% quartile; Q3, 75% quartile. TB, tuberculosis.

Legend: *Unless otherwise specified; [†]1 result missing

TABLE 2. Within-individual comparisons of semi-quantitative GeneXpert MTB/RIF Ultra results between sputum and saliva, among those with positive test results on both sample types (n=72).

		Saliva Xpert Results				
Sputum Xpert Results		Trace	Very Low	Low	Moderate	High
Trace		0	0	0	0	0
Very Low		0	0	1	0	0
Low		2	5	6	0	2
Moderate		0	4	15	2	1
High		2	3	20	8	1

Legend: Shading intensity is proportional to the frequency of paired results by semi-quantitative grade across the two sample types.

RESPONSE TO REVIEWERS

Editors' Comments

Comments:

C1. Abstract appears misleading in its characterization of the study population, which seems to imply that all persons being evaluated for TB were included in the study population. When in fact, it was only sputum Xpert positive that were included in this sub-study. By not including those who were sputum Xpert negative (but could have been sputum culture positive), the sensitivity is likely higher than would have been observed had everyone in the parent study been included. The reviewers suggest a revision of the manuscript to describe a feasibility study and minimize any discussion of sensitivity. I agree with them.

We thank the Editor and other reviewers for this important point. We have edited the opening line of the abstract and other portions of the manuscript to clearly specify the study population (top of page 2 of the tracked-changes version):

“The objective of this prospective, observational study carried out at China-Uganda Friendship Hospital-Naguru in Kampala, Uganda, was to determine the performance of GeneXpert MTB/RIF Ultra (Xpert Ultra) molecular testing on saliva for active tuberculosis (TB) disease among consecutive adults undergoing TB diagnostic evaluation who were Xpert Ultra-positive on sputum.”

In addition, as requested by the Editor and Reviewers, we have comprehensively discussed the limitations of our sensitivity estimates throughout the manuscript, including the concerns expressed by the Editor and the Reviewers about the recruitment strategy creating a selection bias by systematically excluding sputum Xpert Ultra-negative, culture-positive individuals. Specifically, we have now carried out additional analyses of the 153 consecutive patients enrolled in the parent study showing that such individuals were only rarely excluded, so rarely that the exclusions do not meaningfully alter our sensitivity estimates. The results of this analysis are summarized in the updated Figure 1 legend below.

*“**Legend:** *Thirty-eight of 40 patients who were excluded for being sputum Xpert Ultra-negative were also composite sputum mycobacterial culture-negative, leaving only two sputum Xpert Ultra-negative patients who were culture-positive. †Thirteen of 15 patients excluded because sputum Xpert Ultra testing was not done were sputum microscopy-positive and sputum culture-positive, while two were sputum culture-indeterminate.”*

The smear-positive patients would all be expected to have been positive by sputum Xpert Ultra, suggesting that sputum Xpert Ultra and composite sputum mycobacterial culture results would have been concordant in the 15 patients in whom sputum Xpert Ultra testing was not done. Thus, out of the 153 patients, we identified only two patients who were composite sputum mycobacterial culture-positive and Xpert Ultra-negative.

We do acknowledge that the generalizability of our sensitivity estimates remains in question given the high smear-positivity rate, as discussed below in our response to Reviewer 3, comment C5.

C2. Additionally, there is have biosafety concerns regarding the lower concentration of Cepheid Sample Reagent used to treat the saliva samples. Sample Reagent does inactivate bacilli in sputum when used at the manufacturer's recommendations. To my knowledge, inactivation rates of lower concentrations of Sample Reagent have not been evaluated in saliva. While this concentration may still provide adequate inactivation, this should be evaluated, or samples should be handled in a biosafety cabinet in order to ensure the safety of lab personnel.

Thank you for this insightful comment, which we have incorporated as a new limitation to the study in the Discussion section on page 13, near the top:

RESPONSE TO REVIEWERS

“Fourth, we did not evaluate the optimal sample reagent(SR)-to-saliva ratio. Unlike sputum, saliva may not require processing with a mucolytic. Therefore, reducing the SR-to-saliva ratio from the standard 2:1 ratio to a 1:1 ratio, as we did, could improve analytic sensitivity by minimizing dilution of target DNA. However, SR also plays an important role in sterilizing mycobacteria, and reducing the final concentration of SR could create a biosafety risk if processing occurs outside a Level II biosafety cabinet, as may commonly occur in low- and middle-income settings. Both the risk of aerosolizing Mtb from sputum during Xpert processing and the effectiveness of SR in sterilizing sputum at ratios $\geq 2:1$ have been well-documented in sputum [27-29], but not in saliva. Future studies should assess the aerosolization risk in saliva and assess the sterilizing effectiveness of SR in saliva at different dilutions.”

Reviewer #1 (Comments for the Author):

C1. This manuscript reports the important finding that saliva samples analyzed by GeneXpert Ultra may serve as a non-sputum alternative for TB diagnosis. Identifying viable non-sputum samples is critical to the fight against TB, and it can greatly benefit the patients and healthcare workers.

I would consider the study presented in this manuscript to be a successful proof-of-concept study, where the population and methods gave the saliva samples the best chance to perform well relative to sputum testing. It is certainly key step in establishing saliva as a diagnostic sample, though additional research will be necessary before saliva is more widely collected and used.

Specifically, the following points should be addressed:

We thank Reviewer 1 for the positive reception and for recognizing the importance of the findings in the context of the proof-of-concept study design. Below, we have addressed each of Reviewer 1’s comments:

C2. Line 20: Replace "of" with "for". **Done.**

C3. Line 22: Delete "for example". **Done.**

C4. Line 24: Add specific sensitivity data for the study referenced (Reference 6). **Added.**

C5. Lines 28-29: A citation is needed here. **We have added a citation but also modified this text because the original work referenced has been superseded by a larger and more recent study from the same group.**

C6. Line 56: "Speciation" is spelled incorrectly. **Corrected.**

C7. Lines 62-64: By collecting saliva after sputum, it is possible that the bacilli from the sputum remained in the mouth. I'd like this addressed as a caveat/limitation of this study and for follow-up studies to collect saliva before sputum (or at least compare before vs after).

We understand Reviewer 1’s concern and have added this insight to the Discussion as an additional study limitation, incorporating additional hypotheses for evaluation as suggested by other Reviewers. It appears on page 15, at the end of the second-to-last paragraph:

“Last, because saliva was collected after sputum, residual bacilli left in the mouth after coughing may have exaggerated the diagnostic yield of saliva collected without cough. Alternatively, expectoration may have depleted the mouth of Mtb bacilli, reducing sensitivity. Future studies should compare the yield with and without coughing before saliva collection.”

RESPONSE TO REVIEWERS

C8. Line 118: Saliva Xpert sensitivity relative to sputum Xpert, and sputum Xpert sensitivity relative to the combined sputum culture reference standard should also be calculated and included here.

Thank you for this suggestion, which we have implemented in the Results at the top of page 9. We now report both saliva Xpert Ultra sensitivity and sputum Xpert Ultra sensitivity relative to a primary reference standard of composite sputum culture, and also relative to a secondary reference standard of sputum Xpert Ultra.

“Seventy of the 78 patients with sputum culture-confirmed TB were salivary Xpert Ultra positive, giving an overall sensitivity of 90% (95% CI 81-96%) relative to the composite sputum culture-based reference standard. Seventy-two of the 81 sputum Xpert Ultra-positive patients were salivary Xpert Ultra-positive, giving an overall sensitivity of Xpert Ultra on saliva of 89% (95% CI 80-95%) relative to the sputum molecular reference standard. In comparison, the composite sensitivity of fluorescence smear microscopy (FM) on two sputa was 87% (95% CI 77-94%) relative to the sputum culture-based reference standard, and 84% (95% CI 74-91%) relative to the sputum molecular reference standard. The sensitivity of sputum Xpert Ultra was 100% (95% CI 95-100%) relative to the sputum culture-based reference standard and 100% (95% CI 96-100%) relative to the sputum molecular reference standard.”

C9. Line 153: Clarify what "sensitivity of 98%" refers to. Is it the PCR assay or the diagnostic sensitivity of saliva in reference 15? If it is the diagnostic sensitivity of saliva, what is it relative to? Sputum?

We have revised the sentence to clarify that we are referring to diagnostic sensitivity, in this case the proportion of confirmed TB cases who were TB positive on saliva PCR testing (note that the cited article does not specify the reference standard used to confirm the TB):

“The use of saliva for molecular diagnosis of TB was first described in a convenience sample of 52 adult TB patients in Japan in whom testing of saliva using a lab-developed, nested PCR assay targeting the 65 kD mycobacterial antigen detected TB in 98% of those with confirmed TB [19].”

C10. Line 155: When you state "very low sensitivity of 39% for TB testing [16]" what sample/test is this relative to? And were the samples tested in reference 16 analyzed by GeneXpert or manual PCR?

These saliva samples were tested by sputum Xpert Ultra and the sensitivity was determined relative to mycobacterial culture. We have now added this information:

“In a more recent study of 44 sputum smear- and culture-positive TB patients, including 35 in South Africa and 9 in South Korea, Shenai et al reported that saliva tested with Xpert Ultra had a very low sensitivity of 39% for active TB disease relative to a reference standard of sputum liquid mycobacterial culture [19].”

C11. Lines 170-173: I recommend referencing a more recent oral swab study, such as Luabeya et al 2018, or Wood/Andama et al, 2021.

We now reference the more recent Wood/Andama publication.

C12. Line 192: Reference 23 doesn't test saliva, so it is not relevant. It should be removed.

We understand the reviewer’s concern and have revised the text to clearly specify that the citation does not include saliva. We have chosen to retain this information (a Cochrane systematic review of extra-pulmonary specimens for TB) to show that the Xpert assay has high analytic specificity across a range of body fluid types (not only sputum), while clarifying some of the reasons why specificity might be lower in saliva, which we feel may help guide future research on the specificity of saliva Xpert.

RESPONSE TO REVIEWERS

“A recent systematic review found that pre-Ultra Xpert MTB/RIF and/or Xpert Ultra assays have a high specificity on a variety of body fluid types, including pleural fluid, peritoneal fluid, pericardial fluid, lymph node aspirates, bone or joint aspirates, urine, and blood, although no studies of saliva Xpert were identified [26]. Studies of saliva Xpert Ultra specificity are now needed, because the oral cavity is in direct contact with environmental air, increasing the risk of false positive results in high-transmission environments including health-care facilities.”

C13. Lines 180-194: To the paragraph on limitations, add that saliva was collected after sputum, which may have biased the results. It could go either way; expectoration could have depleted the available bacilli in the mouth or coughing up sputum could have deposited additional bacilli in the mouth.

Thank you for this insightful comment. We have added this on page 15, at the end of the second-to-last paragraph, as summarized above in our response to Reviewer 1’s earlier comment C7.

C14. Line 197: Saliva Xpert sensitivity relative to sputum Xpert needs to be provided here, or earlier in the manuscript.

Thank you for this suggestion. We have added this to the Results section, under Diagnostic Performance on page 10:

“Seventy-two of the 81 sputum Xpert Ultra-positive patients were salivary Xpert Ultra-positive, giving an overall sensitivity of Xpert Ultra on saliva of 89% (95% CI 80-95%) relative to the sputum molecular reference standard.”

Reviewer #3 (Comments for the Author):

C1. Lines 23-26. The study described in this manuscript does not address the objective described in the first sentence of the abstract. The study does not assess the performance of the test with saliva samples from persons being evaluated for TB. Rather, it assesses the performance of test with saliva samples from persons being evaluated for TB who have produce sputum specimens that are Xpert positive. These are very different populations of test subjects.

Thank you. This has now been changed in the Introduction and Abstract sections, as detailed above in our response to the Editor, comment C1.

C2. Lines 28-29 and lines 155-157. The sensitivity compared to culture was 93.6% (73/78), not 90%.

Thank you for identifying this miscalculation. The number of salivary Xpert positives was actually 70 of 78 (not 73 as written earlier). This now been corrected in the Results and in the Abstract, we have modified Figure 1 to show these details.

“Seventy of the 78 patients with sputum culture-confirmed TB were salivary Xpert Ultra positive, giving an overall sensitivity of 90% (95% CI 81-95%) relative to the composite sputum culture-based reference standard.”

C3. Lines 32 to 34 (also lines 162-164). The sensitivity was 46% higher (not lower) in smear-positive vs smear-negative.

Thank you, we have corrected this in the Results and in the Abstract.

RESPONSE TO REVIEWERS

C4. A potential confounding aspect of the study design is that the saliva samples were collected after the sputum samples were collected. It is not obvious that waiting 2 hours is sufficient time to avoid potential 'contamination' of the saliva samples with bacteria introduced into the mouth during sputum collection.

Yes, we do understand the Reviewer's concern. As detailed above in response to Reviewer 1, comments C7 and C13, this has been added to the limitations of the study.

C5. The authors should discuss how the data from this somewhat unusual population of test subjects can be extrapolated to the typical population of persons being evaluated for TB. That is, in the overall pool of recruited test subjects, more than half were bacteriologically confirmed TB patients and almost half were smear positive. This pool of test subjects was further restricted to Xpert-positive test subjects, 87% of whom were smear positive.

Thank you for this comment. We have discussed this issue as the first limitation of the Discussion on page 14, in the first full paragraph:

“First, our study population included an exceptionally high proportion of sputum smear-positive individuals. In addition, because our primary study objective was to evaluate feasibility and preliminary sensitivity, we did not include patients with non-productive cough or children, two ideal target populations for salivary testing who are likely to have more paucibacillary disease. Diagnostic sensitivity is likely to be lower in paucibacillary populations, as suggested by the lower sensitivity that we observed among sputum smear-negative individuals and among persons living with HIV. However, in the current study, we found that even though saliva is more paucibacillary than sputum as assessed by Xpert Ultra's semi-quantitative measurement of mycobacterial load, diagnostic sensitivity was similar between the two specimen types, likely because of the extremely low threshold of detection and high analytic sensitivity of the Xpert Ultra assay [11].”

We have addressed the concern about restriction to Xpert-positive test subjects in our above response to the Editor's comment C1.

C6. Lines 177-178, The 'rigorously defined reference standard' should be clearly stated.

This is a good suggestion, which we have implemented in the Discussion at the top of page 14:

“...we showed that diagnosis of TB using Xpert Ultra on saliva is feasible and had a high sensitivity relative to a rigorously defined reference standard of composite sputum mycobacterial culture.”

C7. Line 195. 44 smear-negative samples? Line 198 suggests it should be 44 smear-positive samples.

We apologize for the ambiguity – we meant smear-and-culture-positive samples but have now corrected this as follows:

“...44 sputum smear-positive and culture-positive TB patients...”

C8. GeneXpert refers to the instrument. The test is the Xpert MTB/RIF Ultra test. The authors should take care to refer to the test in such a way as to avoid confusion between the Xpert MTB/RIF test and the Xpert MTB/RIF Ultra test.

Yes, thank you for pointing this out. We have defined and adopted the shorthand phrases “Xpert Ultra” and “pre-Ultra Xpert MTB/RIF” to ensure clarity throughout the manuscript.

RESPONSE TO REVIEWERS

Reviewer #4 (Public repository details (Required)):

Authors indicate in Lines 133-134 that a dataset will be provided prior to publication.

We have added the following citation at the top of page 9:

“A comprehensive, de-identified dataset containing individual-level data is publicly available for download [17].

17. Byanyima, P. *Feasibility and sensitivity of saliva GeneXpert MTBRIF Ultra for tuberculosis diagnosis in Ugandan adults: Dataset*. 2022; Available from: <https://doi.org/10.5061/dryad.2jm63xsrq>.

Reviewer #4 (Comments for the Author):

Introduction

C1. Thank you for the opportunity to review this manuscript. The authors have provided evidence that saliva may be a suitable specimen for the diagnosis of TB, while acknowledging that future studies are needed in order to address lingering questions. The authors collected saliva in a population of sputum Xpert positive patients and showed that there were similar rates of detection of tuberculosis in saliva Xpert Ultra as compared to sputum smear microscopy using sputum culture as the reference standard. While a population of sputum Xpert positive patients is convenient, it does raise the concern that such a population is more likely to test positive with saliva Xpert Ultra as well as sputum culture, thus inflating the sensitivity higher than would be observed in a sputum Xpert negative population.

Thank you for this excellent point, which we have addressed above in our response to the Editor’s comment C1.

C2. Saliva samples were treated at a 1:1 ratio with Cepheid's Sample Reagent. There are potential biosafety considerations using this lower concentration of Sample Reagent that the authors did not address in this study. The authors are encouraged to comment on biosafety concerns as well as temper their conclusions given the selective population in this study.

Thank you for this insightful comment, which we have addressed above in our response to the Editor’s comment C2.

Major Comments

C3. The authors chose to treat the saliva samples with Cepheid Sample Reagent at a 1:1 ratio, which is used for some other non-pulmonary samples such as cerebrospinal fluid. In contrast, sputum is treated at a 2:1 Sample Reagent to sputum ratio. As reported, other groups, such as the South Africa/South Korea study (1) had treated saliva at a 2:1 ratio. The purpose of the sample reagent is two-fold, to liquefy viscous samples, as well as to render bacilli unviable and thus lowering the biohazard risk. As reported in Helb et al. (2) killing assays in spiked sputum were utilizing two volumes of Sample Reagent per volume of sputum. That group showed that at that concentration, after 15 minutes, viability was reduced by at least 8-logs in sputum. This is an important consideration in the safety of laboratory staff to ensure that they are not unnecessarily exposed to viable bacilli. To the reviewer's knowledge, no such killing assays have been conducted for saliva samples at a lower Sample Reagent concentration. If samples are handled within a biosafety cabinet at all times, concerns regarding a lower Sample Reagent concentration are alleviated; however, if treated samples are opened on an open bench, lab workers could potentially be exposed to viable bacilli. The authors are encouraged to specify that samples should be handled within a biosafety cabinet at all times or conduct killing assays to show that lower concentrations of Sample Reagent still result in an adequate reduction in bacilli viability.

RESPONSE TO REVIEWERS

As noted in our response to the Editor's comment C2, we have added a limitation about the biosafety concerns of treating saliva at a ratio of 1:1, including the importance of using a biosafety cabinet. We have also highlighted the need for additional studies of the risks and benefits of different SR:saliva dilutions.

C4. In the Abstract, the authors state, "The objective... was to determine the performance of GeneXpert MTB/RIF Ultra (Xpert) testing on saliva for active tuberculosis (TB) disease among consecutive adults undergoing diagnostic evaluation." This statement is misleading. While the parent study did enroll consecutive adults undergoing TB evaluation, in the sub-study, saliva samples were only collected among adults that had already tested positive for tuberculosis using sputum Xpert. This has important implications in the interpretation of the reported sensitivity. As is noted in the Discussion regarding Wood et al. (3), case-control study designs are prone to inflate diagnostic accuracy. The authors of this study do not acknowledge this same limitation in their study design, which is effectively a "case-only" study population. The authors do note the limitation of not being able to determine specificity, but do not note the limitation of inflated sensitivity. The authors are encouraged to acknowledge this major limitation throughout the manuscript.

Thank you for this insightful point, which highlights the issue of selection bias in case-only diagnostic studies. We have added this as an additional limitation on page 14 of the Discussion, along with additional analyses summarized in our above response to the Editor's comment C2:

"Although case-only studies are more cost-efficient, they have the limitation of potentially inflating sensitivity by excluding patients who would have been diagnosed with TB by a more sensitive inclusion criterion, such as sputum mycobacterial culture or clinical evaluation [25]. However, we found that only two of the 153 patients screened for this study were likely to be sputum Xpert Ultra-negative and culture-positive, making the selection bias too small to meaningfully influence our sensitivity estimates."

C5. Saliva was collected "at least two hours after sputum collection." Is this a sufficient period of waiting? Why was saliva not collected before sputum? It could be reasonably assumed that the process of expectorating sputum would leave some bacilli in the mouth that could then later be released in saliva. Had the order of collection been reversed (saliva first and then sputum), one might expect detection in saliva Xpert to be lower. The authors are strongly encouraged to discuss the rationale for this methodology, and discuss whether future studies should explore sample collection order.

1) Shenai S, Amisano D, Ronacher K, et al. Exploring alternative biomaterials for diagnosis of pulmonary tuberculosis in HIV-negative patients by use of the GeneXpert MTB/RIF assay. *J Clin Microbiol.* 2013;51(12):4161-4166. doi:10.1128/JCM.01743-13

2) Helb D, Jones M, Story E, et al. Rapid Detection of Mycobacterium tuberculosis and Rifampin Resistance by Use of On-Demand, Near-Patient Technology. *J Clin Microbiol.* 2010;48(1):229-237. doi:10.1128/JCM.01463-09

3) Wood RC, Luabeya AK, Weigel KM, et al. Detection of Mycobacterium tuberculosis DNA on the oral mucosa of tuberculosis patients. *Sci Rep.* 2015 Mar 2;5:8668. doi:10.1038/srep08668

We appreciate these citations and references. As noted in our prior response to Reviewer 1 (C7 and C13) and Reviewer 3 (C4), we have added this as a limitation in the Discussion on page 15. We have also included the suggested citations:

7. Wood, R.C., et al., *Detection of Mycobacterium tuberculosis DNA on the oral mucosa of tuberculosis patients. Scientific reports, 2015. 5: p. 8668-8668.*

19. Shenai, S., et al., *Exploring Alternative Biomaterials for Diagnosis of Pulmonary Tuberculosis in HIV-Negative Patients by Use of the GeneXpert MTB/RIF Assay. Journal of Clinical Microbiology, 2013. 51(12): p. 4161-4166.*

RESPONSE TO REVIEWERS

27. Helb, D., et al., *Rapid detection of Mycobacterium tuberculosis and rifampin resistance by use of on-demand, near-patient technology. J Clin Microbiol, 2010. 48(1): p. 229-37.*

Minor Comments

C6. Line 102: The authors are encouraged to explain the instructions provided to patients for collecting saliva samples in greater detail to ensure that other groups are able to recreate sampling as closely as possible.

Yes, this is important. We have expanded our description of the saliva collection method in the Methods section on page 7:

“After sputum collection, participants were asked not to eat or drink before saliva collection. At least two hours after sputum collection, they were then instructed to deposit ≥ 1 mL of saliva in a sterile 50 mL conical specimen cup, taking care not to intentionally cough before saliva collection. Saliva was immediately transferred to the laboratory for Xpert Ultra testing.”

C7. Table 2: The authors are encouraged to include a negative row and column to show those individuals positive only in saliva Xpert and those positive only in sputum Xpert.

Thank you for this suggestion, we have added this to Table 2.

Staff Comments:

Preparing Revision Guidelines

For complete guidelines on revision requirements, please see the journal Submission and Review Process requirements at <https://journals.asm.org/journal/Spectrum/submission-review-process>. Submissions of a paper that does not conform to Microbiology Spectrum guidelines will delay acceptance of your manuscript. "

Please return the manuscript within 60 days; if you cannot complete the modification within this time period, please contact me. If you do not wish to modify the manuscript and prefer to submit it to another journal, please notify me of your decision immediately so that the manuscript may be formally withdrawn from consideration by Microbiology Spectrum.

If your manuscript is accepted for publication, you will be contacted separately about payment when the proofs are issued; please follow the instructions in that e-mail. Arrangements for payment must be made before your

RESPONSE TO REVIEWERS

article is published. For a complete list of Publication Fees, including supplemental material costs, please visit our website.

August 30, 2022

Dr. J. Lucian Davis
Yale School of Public Health
New Haven, CT

Re: Spectrum00860-22R1 (Feasibility and Sensitivity of Saliva GeneXpert MTB/RIF Ultra for Tuberculosis Diagnosis in Adults in Uganda)

Dear Dr. J. Lucian Davis:

Your manuscript has been accepted, and I am forwarding it to the ASM Journals Department for publication. You will be notified when your proofs are ready to be viewed.

Sincerely,

Rita Oladele
Editor, Microbiology Spectrum
